# MyBrain-Seq: A Pipeline for MiRNA-Seq Data Analysis in Neuropsychiatric Disorders

**DOI:** 10.3390/biomedicines11041230

**Published:** 2023-04-21

**Authors:** Daniel Pérez-Rodríguez, Roberto Carlos Agís-Balboa, Hugo López-Fernández

**Affiliations:** 1Neuro Epigenetics Lab, Health Research Institute of Santiago de Compostela (IDIS), Santiago University Hospital Complex, 15706 Santiago de Compostela, Spain; 2Translational Neuroscience Group, Galicia Sur Health Research Institute (IIS Galicia Sur), Área Sanitaria de Vigo-Hospital Álvaro Cunqueiro, SERGAS-UVIGO, CIBERSAM-ISCIII, 36213 Vigo, Spain; 3Translational Research in Neurological Diseases Group, Health Research Institute of Santiago de Compostela (IDIS), Santiago University Hospital Complex, 15706 Santiago de Compostela, Spain; 4Servicio de Neurología, Hospital Clínico Universitario de Santiago, 15706 Santiago de Compostela, Spain; 5CINBIO, Department of Computer Science, ESEI-Escuela Superior de Ingeniería Informática, Universidade de Vigo, 32004 Ourense, Spain; hlfernandez@uvigo.es; 6SING Research Group, Galicia Sur Health Research Institute (IIS Galicia Sur), SERGAS-UVIGO, 36213 Vigo, Spain

**Keywords:** miRNA-Seq, miRNAs, differential expression, functional analysis, neuropsychiatry, reproducibility, myBrain-Seq, Compi, NGS, epigenetics

## Abstract

High-throughput sequencing of small RNA molecules such as microRNAs (miRNAs) has become a widely used approach for studying gene expression and regulation. However, analyzing miRNA-Seq data can be challenging because it requires multiple steps, from quality control and preprocessing to differential expression and pathway-enrichment analyses, with many tools and databases available for each step. Furthermore, reproducibility of the analysis pipeline is crucial to ensure that the results are accurate and reliable. Here, we present myBrain-Seq, a comprehensive and reproducible pipeline for analyzing miRNA-Seq data that incorporates miRNA-specific solutions at each step of the analysis. The pipeline was designed to be flexible and user-friendly, allowing researchers with different levels of expertise to perform the analysis in a standardized and reproducible manner, using the most common and widely used tools for each step. In this work, we describe the implementation of myBrain-Seq and demonstrate its capacity to consistently and reproducibly identify differentially expressed miRNAs and enriched pathways by applying it to a real case study in which we compared schizophrenia patients who responded to medication with treatment-resistant schizophrenia patients to obtain a 16-miRNA treatment-resistant schizophrenia profile.

## 1. Introduction

The rapid development in the field of transcriptomics has allowed the existing knowledge about the molecular mechanisms of pathogenesis to be expanded. Over the last two decades, RNA-Seq technology has been used in translational medicine as a valuable tool for disease profiling and biomarker identification. This has led to important discoveries [1,2,3] and the foundation of precious resources such as ENCODE [4], the Cancer Genome Atlas [5] or the pathway database Reactome [6]. RNA-Seq technology has also allowed a new approach to study some of the most heterogeneous disorders, such as many neuropsychiatric conditions. The study of gene-expression regulators such as microRNAs (miRNAs) has offered a new angle from which neuropsychiatric conditions could be understood: the environmental influence on gene expression and the different responses to that environment. This application of the RNA-Seq technology is commonly known as miRNA-Seq.

MiRNAs are short non-coding RNA molecules involved in mRNA silencing. As epigenetic regulators, they are closely associated with adaptation processes and their expression levels are affected by events such as diet, sleep, stress or medications [7]. Nowadays, miRNAs are recognized as important etiological factors of neuropsychiatric diseases such as schizophrenia [8,9], depression [10,11] or Alzheimer’s disease [12,13], and thus have potential to be biomarkers and therapeutic targets. As a result, the number of studies using miRNA-Seq has increased in the last decade [7], leading to the creation of miRNA databases such as the human microRNA disease database (HMDD) [14], the Central Nervous System microRNA Profiles (CNS microRNA Profiles) [15] and the Human miRNA Expression Database (miRmine) [16].

However, the extensive use of the miRNA-Seq technology, together with the absence of a standardized methodology for bioinformatics analysis, has raised concerns about its reproducibility [17]. The main concerns are the great variability in the analysis procedures [18,19], the large influence that biological references have on the results [19,20], the application of generic RNA methods to miRNA data [21] and the biological variability itself, that makes the comparison between studies complex [22,23].

Many bioinformatic tools were designed to perform specific tasks in a miRNA-Seq analysis. These tasks range from miRNA identification, such as in the case of Mirnovo [24], UEA sRNA workbench [25] or miRDeep [26], to annotation tools as miRBase [27], Rfam [28] or miRIAD [29], to interaction databases such as Diana Tarbase [30], starBase [31] or PceRBase [32]. Importantly, few tools integrate these solutions into a single pipeline aiming for a complete miRNA-Seq analysis [33]. Some pipelines such as the ENCODE miRNA-Seq pipeline [34] or miRge3.0 [35] process the raw data until the transcript annotation and quantification step. Others, such as CAP-miRSeq [36], process the data and perform a differential expression analysis. Finally, very few pipelines perform a target annotation and a functional analysis, as in the case of miARma-Seq [37]. To the best of our knowledge, there are neither bioinformatic pipelines with a functional analysis corrected for the enrichment bias of the miRNA annotations [7,21] nor a pipeline that generates a network of miRNA–protein molecular interactions.

In this context, we present myBrain-Seq (https://github.com/sing-group/my-brain-seq), a highly modular pipeline for performing replicable miRNA-Seq analysis, from the preprocessing of the raw data to the generation of a network of miRNA–protein interactions. It covers most of the necessities of a miRNA-Seq study of neuropsychiatric data, which were identified by thoroughly reviewing current miRNA-Seq methodologies in the field [7] and translating that information into the pipeline design [17,38]. Finally, myBrain-Seq uses Docker technology, which ensures a stable environment regarding the dependencies and biological annotations used for data analysis, thus enhancing the replicability of the whole process [17,38].

In this study, we demonstrate the usefulness of myBrain-Seq by showing how it is applied to a real case study in which we compared schizophrenia patients who responded to medication with treatment-resistant schizophrenia patients to obtain a 16-miRNA treatment-resistant schizophrenia profile [8].

## 2. Materials and Methods

### 2.1. MyBrain-Seq, a Pipeline for miRNA-Seq Analysis

MyBrain-Seq is a Compi [39,40] pipeline to automatically analyze miRNA-Seq data in a highly reproducible way. It helps to find a profile of differentially expressed miRNAs between two conditions, assesses its potential classification power using hierarchical clustering analysis and aids in the discovery of biological pathways potentially affected by the conditions. It also helps to discover limitations in the quality of the data that may affect the conclusions of the study.

As depicted in Figure 1, the workflow entails the following main steps: (i) preprocessing, (ii) expression analysis, (iii) hierarchical clustering, (iv) functional analysis and (v) network analysis. Each of these steps comprises several tasks. The first step, preprocessing, includes quality control of the raw data, removal of adapter sequences, alignment to a reference genome, quality control of the alignments and annotation and quantification of the aligned transcripts. The second step, expression analysis, includes two differential expression analyses (DEA) performed using two different software packages as well as the integration of both results using a custom myBrain-Seq script. Finally, the third, fourth and fifth steps are single-task steps specifically designed for myBrain-Seq. The following subsections provide more details about each one.

### 2.2. Quality Control and Adapter Removal

The first two preprocessing steps (Figure 1, step 1, a & b) are to discover samples with low-quality sequences as well as to detect the presence of foreign DNA contamination such as sequencing adapters. An evaluation of the results of this module is important for discarding samples from the downstream analysis and therefore it is the first step to be performed.

On one hand, myBrain-Seq includes the analysis of sequence quality scores, sequence length distribution, overrepresented sequences, adapter content and other parameters included in the FastQC tool [41]. On the other hand, the adapter trimming is performed using the Cutadapt tool [42]. Both quality control and adapter trimming analyze several samples at once, thus accelerating the analysis of large sample sizes.

### 2.3. Alignment to the Reference Genome

Alignment is the process of mapping reads against a biological reference (usually a genome or transcriptome) in order to assign genomic positions to each of the sequences of the raw data. The alignment in myBrain-Seq (Figure 1, step 1, c–e) encompasses building of a genome index (if needed), the alignment of the reads to a reference genome, a file format conversion of the results and a quality control of the alignments. First, myBrain-Seq performs the alignments with the Bowtie 1 tool [43], a short sequence aligner specializing in mapping short transcripts (such as miRNAs) to large genomes [7]. Bowtie uses a Burrows–Wheeler index of the genome to speed the mapping process and reduce the impact on computer memory. This small memory footprint is used by myBrain-Seq to parallelize the alignments of several samples to accelerate the processing of large batches of files. Next, a file format conversion from Sequence Alignment Map (SAM) to Binary Alignment Map (BAM) is performed with SAMtools [44]. Finally, a quality control with SAMtools stats and SAMtools bcftools [45] brings information about the sequencing depth of the samples, an important parameter that allows an estimate of the reliability of the RNA-Seq data. Other parameters such as ACGT cycles, sequencing coverage and GC content are also reported.

### 2.4. Transcripts Annotation and Quantification

Aligned sequences in BAM are then mapped to a reference transcriptome. This process, usually known as “annotation”, uses the genomic coordinates of an annotation file to convert the genomic locations of the aligned samples into transcript IDs. Those transcripts are then grouped by ID and quantified in a process known as read summarization or quantification. MyBrain-Seq performs both processes, annotation and quantification, using the software FeatureCounts [46] (Figure 1, step 1, f). It also requires a user-uploaded GTF/GFF file as the biological reference for annotations. Results of the quantification are presented as plain transcript counts.

### 2.5. Differential Expression Analysis

A differential expression analysis (DEA) is the process of finding statistically significant differences in the transcript expression between two different conditions. Those differences can be potentially related to biological alterations and usually are the starting point of the functional analysis and the interpretation of results. MyBrain-Seq uses two well-known software packages for DEA, namely DESeq2 [47] and EdgeR [48] (Figure 1, step 2, a & b), and implements a volcano plot for the visualization of the differentially expressed miRNAs (DE miRNAs). Additionally, myBrain-Seq implements an option to integrate the results of both pieces of software (“integrated results” henceforth), thus offering a more conservative analysis of the data (Figure 1, step 2, c).

Regarding the software for differential expression analysis, DESeq2 normalizes the counts using the median of ratios method [49] prior to the DEA using a negative binomial distribution. On the other hand, EdgeR uses a weighted trimmed mean of the log expression ratios between the samples for the normalization (TMM method) [50] and an exact test for calculating the statistical differences in the miRNA expression. Both software packages apply the Benjamini–Hochberg FDR correction to the *p*-values [51]. MyBrain-Seq is also able to adjust the DEA model for covariates using the user input (more details in the Results section).

The integration of the DESeq2 and EdgeR results is performed by finding their common miRNAs; then, for each of those miRNAs, FDR and *p*-values are averaged. Finally, FDR and log_2_ FC thresholds are set (default FDR < 0.05; |log_2_ FC| ≥ 0.5) to obtain the list of DE miRNAs. The user can manually adjust both of these thresholds as well as the ones used for subsequent tasks (refer to myBrain-Seq documentation). Additionally, a Venn diagram is created to offer a visual representation of the coincidences: first DESeq2 and EdgeR results are filtered by FDR (default FDR < 0.05; |log_2_ FC| ≥ 0.5), then coincidences and differences are counted and plotted using the R package “VennDiagram” [52]. Volcano plots of all the results are created using the R package “EnhancedVolcano” [53].

### 2.6. Hierarchical Clustering

In the hierarchical clustering step, samples are assigned into clusters using the expression of the DE miRNAs. Samples with similar expression levels will group closely. This grouping can ease the identification of unknown relationships between the data as well as being an estimation of the classificatory power of the DE miRNAs. MyBrain-Seq automatically performs a hierarchical clustering analysis after the DEA using the R package “hclust” [54] (Figure 1, step 3). The whole process is divided into two steps: first, the data are prepared for the hierarchical clustering; second, the hierarchical clustering and figures are generated. The first step, preparation of the data, comprises the optional normalization of all the samples with DESeq2 [47] (flag -deseqNormalizationHclust) and the filtering of the DESeq2 and/or EdgeR results by FDR and log_2_ FC (default FDR ≤ 0.05; |log_2_ FC| ≥ 0.5) to obtain the DE miRNAs. After that, a table is built with one column per sample, each column being the DE miRNAs counts for a specific sample. In the second step, the counts are scaled using the R “scale” function, the matrix of Euclidean distances between samples is created and the clustering is performed using the “ward.D2” method of the R package “hclust” [54]. Finally, a dendrogram and a heatmap are generated using the R packages “dendextend” [55] and the function “heatmap.2” of the package gplots [56], respectively.

### 2.7. Functional Analysis

A functional analysis aims to put the differences spotted in the DEA into a biological context, suggesting biological pathways that might be useful for the investigator to analyze. The functional enrichment analysis of myBrain-Seq (Figure 1, step 4) uses two annotation sources which are included in the myBrain-Seq Docker image: the miRNA-gene annotations from the Diana TarBase [30] and the gene-pathway annotations of the Reactome databases [6]. We selected Diana TarBase as our target annotation database because it contains only miRNA–target interactions that have been experimentally validated. Additionally, both Diana TarBase and Reactome are publicly accessible and updated on a regular basis. Regarding the enrichment analysis, myBrain-Seq follows the strategy proposed by Godard and van Eyll [21] to be specific for miRNA data, briefly: (i) using TarBase annotations, protein coding genes in Reactome pathways are converted into lists of miRNAs that target at least one of these genes; (ii) enrichment analysis is performed by comparing DE miRNAs of DESeq2, EdgeR or integrated results to the lists of miRNAs previously associated with the different pathways. Enrichment scores are calculated using a Fisher hypergeometric test. Finally, a word analysis on the enriched terms is performed using the R package “tidytext” [57] and the results are summarized in a figure and presented along with the pathway-enriched table.

Conventional enrichment analysis is performed on genes rather than miRNAs. In a miRNA-Seq analysis, genes are indirectly selected by target prediction; therefore, genes with more targets have more chances to be selected. The consequence of this bias is a non-specific result, usually identifying as enriched the most studied pathways such as cancer-related or generic signaling pathways [21]. MyBrain-Seq enrichment analysis deals with this bias as each miRNA is only represented once in each pathway, thus ensuring the specificity of the results.

### 2.8. MiRNA–Protein Interaction Network

After the functional analysis, the most enriched pathway is used to build a network of miRNA–protein interactions, providing the researcher with a possible molecular context for the observed differences in miRNA expression. The miRNA–protein interaction network step of myBrain-Seq (Figure 1, step 5) uses the same annotation files as in the functional enrichment analysis step plus a protein–protein interaction file from the Reactome database [6]. The network is built by expanding the miRNA–protein interactions present in the most enriched pathway with the Reactome protein–protein interactions. The process is as follows: (i) miRNAs and genes that participate in the most enriched pathway are found using the functional analysis result; (ii) miRNA–protein interactions are found by using the Tarbase annotations file [30]; (iii) protein–protein interactions that participate in the most enrichment pathways are found using the protein–protein interaction file; and (iv) miRNA–protein interactions and protein–protein interactions are merged into a single table. Finally, a table with all the interactions is generated. This table can be easily imported into network analysis software such as Cytoscape [58] for further analysis and expansions. Additionally, an interactive network file in HTML is generated using the R packages “networkD3” [59] and “htmlwidgets” [60].

### 2.9. Summarization of the Quality Controls

Results of the quality control of the samples and the alignment are calculated on a per-sample basis, making overall interpretation of data quality difficult. To avoid this, the last step of myBrain-Seq analysis is the summarization of featureCounts [46], Samtools [45] and FastQC [41] results using MultiQC [61] (see Figure 1). The output of this step is a single HTML report from which different tables and graphs can be generated.

### 2.10. MyBrain-Seq Implementation

MyBrain-Seq is implemented as a Compi pipeline [39,40] and distributed as a Docker image that allows running it effortlessly. All external dependencies (Table 1) are satisfied using Docker images from the pegi3s Bioinformatics Docker images Project [62] (https://pegi3s.github.io/dockerfiles/). The source code of the pipeline is publicly available at GitHub (https://github.com/sing-group/my-brain-seq) under an MIT LICENSE, and the Docker image is available at Docker Hub (https://hub.docker.com/r/singgroup/my-brain-seq) and at Compi Hub (https://www.sing-group.org/compihub/explore/625e719acc1507001943ab7f#overview).

### 2.11. Case Study Dataset: Treatment Resistant Schizophrenia

The application of myBrain-Seq to biomedical research is illustrated in a recent study [8] with the comparison of the miRNA profile of patients with schizophrenia (SZ) and treatment-resistant schizophrenia (TRS). The dataset comprises reads of circulating miRNA of 40 human patients with schizophrenia, of which 19 patients have a normal response to medication (MR; n = 19) and 21 have an insufficient response to medication (MNR; n = 21). The dataset can be downloaded from Gene Expression Omnibus (GEO) under the accession number GSE223043. We also provide a script in Appendix A to automatically download and perform myBrain-Seq analysis on this dataset.

## 3. Results and Discussion

The goal of myBrain-Seq is to offer a modular and highly customizable tool for miRNA-Seq analysis that allows performing replicable studies. It offers a straightforward analysis process that brings together the most common tools in the field embedded in a portable and customizable pipeline. Among myBrain-Seq’s main contributions are the options designed to solve typical problems in the analysis of transcriptomic data such as the high variability of the transcripts abundance (covariate adjustment), bias in the pathway-enrichment analysis resulting from the indirect selection of genes (miRNA-oriented pathway analysis strategy), low replicability of the results (containerized processes, DEA integration) or the need to explore the results in a biological context (miRNA–protein interaction network). The following subsections describe the contributions of myBrain-Seq with more details.

### 3.1. MyBrain-Seq Execution

The sequence of steps needed to start a myBrain-Seq can be described as follows:Creation of the directory tree in the local file system, referred to as “working directory”, shown in Figure 2. The working directory consists of a main directory with two subdirectories: “/input” and “/output”. The input subdirectory is where the parameter files of myBrain-Seq should be placed; the output subdirectory will contain the results after myBrain-Seq execution. This working directory can be initialized using the utilities included in the myBrain-Seq Docker image. This initialization creates a “run.sh” file, used to run the pipeline and templates of the other files required by myBrain-Seq (those inside “/input”). A “README.txt” file is also created with the instructions to fill the template files and run the pipeline.
Figure 2Working directory of myBrain-Seq.
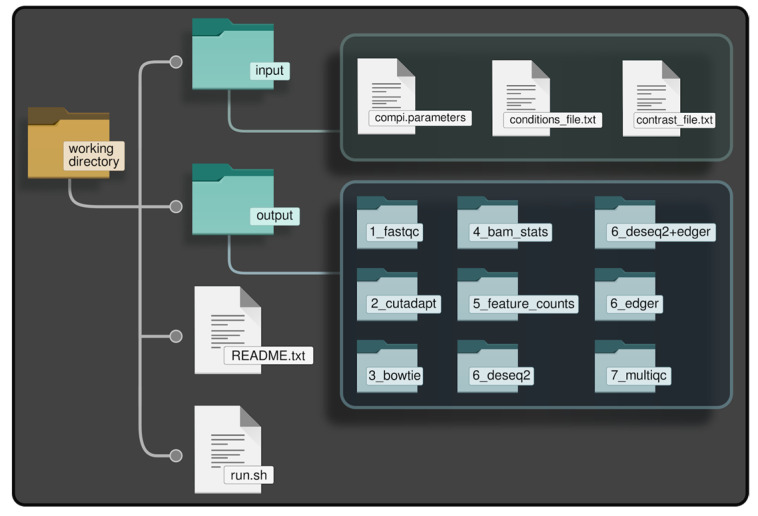

The second step is the preparation of the data. In addition to the FASTQ files, myBrain-Seq needs a reference genome (or Bowtie index) and a GFF file with miRNA annotations as biological references to perform the analysis. It is recommended, but not mandatory, to put all these files inside subdirectories under “/input”. Nevertheless, if they are in other locations (e.g., a shared directory to save disk space), the provided “run.sh” will take care of this and create the appropriate Docker volume bindings in a transparent manner for the user.The third step is the configuration of the analysis. This comprises the creation of three files, namely: “compi.parameters”, “conditions_file.txt” and “contrast_file.txt”. These files are usually placed into the “input” directory.
The “**compi.parameters**” file contains the paths and parameters needed for the analysis, i.e.: path of the working directory, paths to FASTQ files and biological references, paths to “conditions_file.txt” and to “contrast_file.txt” and the adapter sequence. For more information about the optional parameters that can be added, refer to the myBrain-Seq user manual (https://github.com/sing-group/my-brain-seq).The “**conditions_file.txt**” contains the metadata regarding names and conditions of each fastQ file. This file is used by myBrain-Seq to link each sample with a condition and its covariates. Each row of this file contains the name of the FASTQ file, its condition, a user label for that sample and zero or more columns describing the covariates for that sample (e.g., age, sex). All the covariates added in this file will be used in the DEA to adjust the statistical model.The “**contrast_file.txt**” contains the conditions to compare during the analysis and a label for each contrast. Conditions included in this file must be the same as those stated in “contitions_file.txt”. MyBrain-Seq can perform several contrasts in the same pipeline execution if several contrasts are specified in this file, one per line.The final step is running myBrain-Seq analysis using the “run.sh” script created during the working directory initialization (step number 1). This script will use “compi.parameters” as reference, mount all the needed Docker volumes (by extracting the path from the Compi parameters file) and create a directory for the log files of the current execution. MyBrain-Seq users do not need to modify this file, as it is ready to use. Thus, users only need to run the script using the path to “compi.parameters” as the unique argument to start the myBrain-Seq analysis.Both final and intermediate results are saved in the “/output” directory. Such output files are placed in directories corresponding to the different steps of the workflow, namely: “1_fastqc”, “2_cutadapt”, “3_bowtie”, “4_bam_stats”, “5_feature_counts”, “6_deseq2”, “6_deseq2+edger”, “6_edger” and “7_multiqc”. Results from the hierarchical clustering, functional analysis and network analysis are placed in the directories prefixed with “6_”, according to the data from which they were generated. Files from the same contrast are grouped in subdirectories named with the contrast label.

### 3.2. MyBrain-Seq Results

The results of myBrain-Seq are placed in sub-directories inside the “output” directory (see Figure 2). The main results stem from the DEA and are presented for each contrast specified in “contrast_file.txt” and for each DEA method. Figure 3 illustrates the graphical results of a myBrain-Seq analysis:Volcano plot with the results of each DEA; Figure 3A.Venn diagram with the DE miRNA coincidences between DESeq2 and EdgeR; Figure 3B.Dendrogram with the result of the hierarchical clustering; Figure 3C.Heatmap with the result of the hierarchical clustering; Figure 3D.HTML file with a miRNA–protein interaction network of the most enriched pathway; Figure 3E.Lollipop chart with the word frequency of the enriched terms; Figure 3F.

Figure 4 illustrates the tabular results in TSV:Results of the DEA; Figure 4A. Full table in Appendix A.List of DE miRNAs; Figure 4B.Enriched pathways; Figure 4C. Full table in Appendix A.miRNA–protein interaction network; Figure 4D. Full table in Appendix A.

Additionally, myBrain-Seq offers the intermediate files of the analysis to reuse or inspect, namely:Adapter-trimmed FASTQ files.BAM and SAM files resulting from the alignment.A TXT file with the counts of miRNA per sample.A summary of the quantification results.A file per contrast with a subset of counts for that contrast.A TSV file with the expression per sample of each DE miRNA, used for the hierarchical clustering.

Finally, myBrain-Seq generates an HTML file with a summary of the results of the quality control, alignments, assignments and with the quantification of all the samples.

### 3.3. Case Study

An early version of myBrain-Seq (v0.1.0) was used in Pérez-Rodríguez et al. 2023 to perform all analysis steps up to quantification [8]. In this study, myBrain-Seq was used to identify 16 differentially expressed miRNAs (DE miRNAs) between the MR and MNR conditions using DESeq2. The analysis performed can be described using Figure 1, where the reference factor “X” is the schizophrenia condition (MR), the response factor “Y” is the treatment-resistant schizophrenia condition (MNR) and six variables (n = 6) were used to adjust the differential expression model (V1, V2, …, V6). These six variables are: processing bath, sex, drug consumption (alcohol OR tobacco OR illegal), time (hospital arrival/discharge), treatment based on diazepines, oxazepines, thiazepines and oxepins and treatment based on other antipsychotics.

However, there are remarkable differences in the functional analysis and in the miRNA–protein network due to the application of different methodologies applied in both analyses. In Pérez-Rodríguez et al. 2023, we performed a bibliographic search followed by a target prediction to enrich Tarbase [30] annotations. After that, the network was built on Cytoscape [58], expanded and filtered using custom Cytostape [58] filters and StringApp [63]. The pathway enrichment was later performed using the molecules present in the resulting network. On the other hand, as explained before, myBrain-Seq v1.0.0 takes a simpler but effective approach to the functional analysis: to avoid the artificial miRNA target overrepresentation and construct a reduced miRNA–protein network. With this approach, the network is produced after the functional enrichment, thus being smaller and with no overrepresentation biases. This network can be further expanded and filtered externally as we did in Pérez-Rodríguez et al. 2023.

Regarding the enriched pathways, Table 2 offers a comparison between the top ten Reactome-enriched pathways [6] in Pérez-Rodríguez et al. 2023 and myBrain-Seq v1.0.0. In the original study, several pathway databases were used in order to perform the enrichment analysis. On the other hand, myBrain-Seq v1.0.0 only uses Reactome annotations. There are no coincidences between these top ten enriched pathways, probably because StringApp enrichment uses annotations of all levels of the Reactome pathway hierarchy whereas myBrain-Seq uses only the lowest level annotations. This has the advantage of providing more useful results by discarding big unspecific pathways such as “Metabolism of proteins”, “Developmental Biology” or “Disease”, which provide little help in getting to the molecular causes of a condition. Regarding this matter, 217 of the enriched pathways that were discovered using myBrain-Seq v.1.0.0 were also detected in the enriched results of Pérez-Rodríguez et al. 2023 (refer to Appendix A). However, their significance in the context of the entire table differs significantly. For example, the second most enriched pathway in myBrain-Seq, “Activation of anterior HOX genes in hindbrain development during early embryogenesis” (see Table 2), which has a q-value of 1.22 × 10^−5^, is ranked at position 86 and has a q-value of 2.46 × 10^−7^ in the results of Pérez-Rodríguez et al. 2023, which is two orders of magnitude lower (see Appendix A).

In relation to this, it is also worth noting the differences in the scale of the *p* and q values. These differences are likely due to both the lack of specificity of the pathways and the overrepresentation of miRNA targets, resulting in imbalances in the enrichment values (see functional analysis section). Both phenomena produce a high number of false positives, which in turn forces the use of additional filtering strategies for the identification of relevant pathways. Thanks to myBrain-Seq correction, interpretations can be made straight from the enriched table.

## 4. Conclusions

MyBrain-Seq is a highly modular bioinformatics pipeline that specializes in replicable miRNA-Seq data analyses. Created using Compi, it provides a complete set of analyses ranging from raw data preprocessing and DEA to hierarchical clustering, functional analysis and network creation. MyBrain-Seq adaptations to miRNA data include the use of a short-sequence aligner, correction of the DEA model for confounding factors, correction of the artificial target overrepresentation in the functional analysis and creation of a miRNA–protein interaction network.

MyBrain-Seq has already been used in a real case study in which we compared schizophrenia patients who responded to medication to treatment-resistant schizophrenia patients to obtain a 16-miRNA treatment-resistant schizophrenia profile [8]. We were able to reproduce all the findings from the case study up to the functional analysis stage, including the miRNA profile suggested in Pérez-Rodríguez et al. 2023, as well as the hierarchical clustering. By making adjustments to myBrain-Seq functional analysis, we were able to generate more succinct and insightful enriched pathways, while also preventing any biases in the results that could have been caused by an overrepresentation of miRNA targets.

Overall, myBrain-Seq offers a powerful and reliable tool for miRNA-Seq data analysis, which can help researchers identify meaningful biological insights with greater confidence and ease. By reducing the need for additional analysis and providing a complete pipeline for data processing, DEA, hierarchical clustering, functional analysis and network creation, myBrain-Seq can streamline the research process and promote greater reproducibility of the results.

The tool is open for further extension and new features will be included as it is used in new studies. MyBrain-Seq is freely distributed under an MIT license and a complete manual is available at https://github.com/sing-group/my-brain-seq.

## Figures and Tables

**Figure 1 biomedicines-11-01230-f001:**
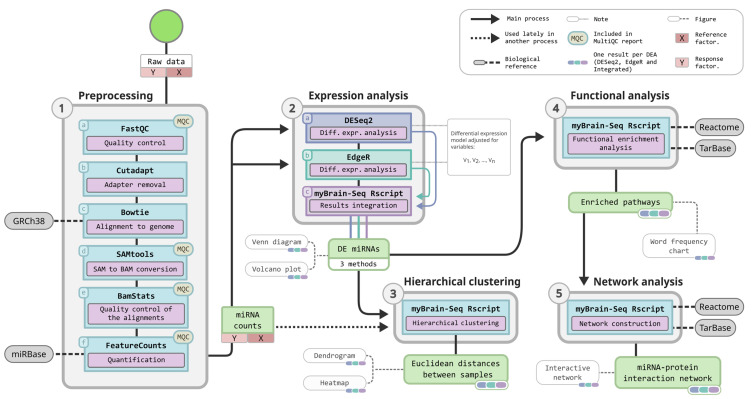
Summary of myBrain-Seq analysis with its five main steps: (1) preprocessing, (2) expression analysis, (3) hierarchical clustering, (4) functional analysis and (5) network analysis. Processes named “myBrain-Seq Rscript” are analysis scripts developed specifically for myBrain-Seq.

**Figure 3 biomedicines-11-01230-f003:**
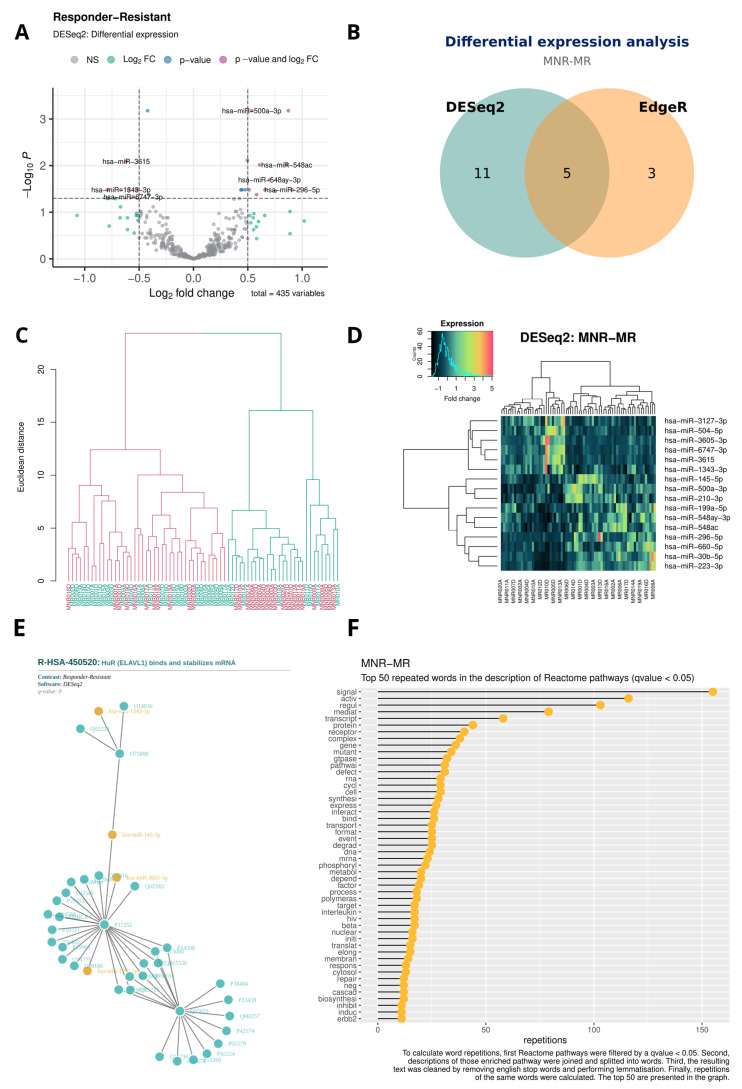
Figures generated after myBrain-Seq analysis: (**A**) Volcano plot of the differential expression analysis; (**B**) Venn diagram comparing the differentially expressed miRNAs resulting from DESeq2 and EdgeR methods; (**C**) dendrogram result of a hierarchical clustering of the samples using the differentially expressed miRNAs; (**D**) heatmap with the levels of expression of the differentially expressed miRNAs per sample; (**E**) miRNA–protein interaction network (portion); (**F**) lollipop chart with the word frequency of the top 50 enriched pathways.

**Figure 4 biomedicines-11-01230-f004:**
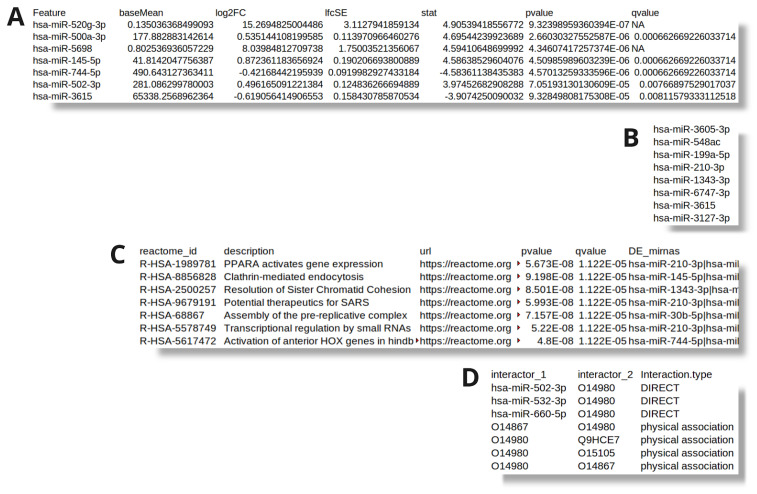
Subset of the main results produced by myBrain-Seq analysis: (**A**) Results of a differential expression analysis for a specific contrast and software (DESeq2, EdgeR or integrated); (**B**) list of differentially expressed miRNAs for a specific contrast and software; (**C**) list of enriched pathways and DE miRNAs implicated in each pathway; (**D**) miRNA–protein interaction table, compatible with network software such as Cytoscape [58].

**Table 1 biomedicines-11-01230-t001:** MyBrain-Seq software dependencies and databases.

Dependencies	Version	Dependencies	Version
pegi3s/r_deseq2	1.32.0	pegi3s/samtools_bcftools	1.10
pegi3s/r_edger	3.36.0	pegi3s/r_data-analysis	4.1.1_v2
pegi3s/r_enhanced-volcano	1.12.0	pegi3s/r_venn-diagram	1.7.0
pegi3s/cutadapt	1.16	pegi3s/r_network	4.1.1_v2_v3
pegi3s/fastqc	0.11.9	pegi3s/multiqc	1.14.0
pegi3s/bowtie1	1.2.3	python3	3.8.5
pegi3s/feature-counts	2.0.0	DIANA Tarbase annotations	8
pegi3s/samtools_bcftools	1.9	Reactome annotations	83

**Table 2 biomedicines-11-01230-t002:** Top 10 enriched pathways in Pérez-Rodríguez et al. 2023 [8] and in the myBrain-Seq reanalysis. Only Reactome pathways [6] were included in this table.

Pérez-Rodríguez et al. 2023 [8]			MyBrain-Seq		
Pathway	*p*-Value	q-Value	Pathway	*p*-Value	q-Value
Metabolism of proteins	2.32 × 10^−55^	5.03 × 10^−52^	HuR (ELAVL1) binds and stabilizes mRNA	4.22 × 10^−8^	1.22 × 10^−5^
Gene expression (Transcription)	1.47 × 10^−54^	1.6 × 10^−51^	Activation of anterior HOX genes in hindbrain development during early embryogenesis	4.80 × 10^−8^	1.22 × 10^−5^
Cellular responses to stress	3.18 × 10^−47^	1.73 × 10^−44^	Transcriptional regulation by small RNAs	5.22 × 10^−8^	1.22 × 10^−5^
Disease	1.08 × 10^−44^	3.89 × 10^−42^	Cyclin E associated events during G1/S transition	5.31 × 10^−8^	1.22 × 10^−5^
Metabolism of RNA	7.31 × 10^−43^	1.99 × 10^−40^	MAPK6/MAPK4 signaling	5.52 × 10^−8^	1.22 × 10^−5^
Cell Cycle	1.33 × 10^−42^	3.22 × 10^−40^	PPARA activates gene expression	5.67 × 10^−8^	1.22 × 10^−5^
Developmental Biology	8.61 × 10^−34^	1.7 × 10^−31^	Cyclin A:Cdk2 associated events at S phase entry	5.68 × 10^−8^	1.22 × 10^−5^
Transcriptional Regulation by TP53	5.17 × 10^−33^	9.37 × 10^−31^	Potential therapeutics for SARS	5.99 × 10^−8^	1.22 × 10^−5^
DNA Repair	2.5 × 10^−32^	4.18 × 10^−30^	Assembly of the pre-replicative complex	7.16 × 10^−8^	1.22 × 10^−5^
Innate Immune System	1.94 × 10^−31^	3.01 × 10^−29^	SUMOylation of ubiquitinylation proteins	8.15 × 10^−8^	1.22 × 10^−5^

## Data Availability

MyBrain-Seq is freely distributed under an MIT license at https://github.com/sing-group/my-brain-seq. The dataset used for the case study is publicly available under Gene Expression Omnibus (GEO) accession number GSE223043.

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
