# Peer review of "MyBrain-Seq: A Pipeline for MiRNA-Seq Data Analysis in Neuropsychiatric Disorders"

_biomedicines, 2023, doi:10.3390/biomedicines11041230_

Round 1
Reviewer 1 Report
In this manuscript, the authors described their miRNA-seq analysis workflow called myBrain-Seq. Each researcher usually shares data analysis workflow through GitHub or as supplementary information of papers. Although it is helpful to prepare a detailed documentation, the authors could do that as part of GitHub document. In this case, there was no significant technological advancement or critical evaluation of the method. There is still much to specify by users, and it is unclear how this analysis workflow contributes to standardization of miRNA analysis.
Author Response
In this manuscript, the authors described their miRNA-seq analysis workflow called myBrain-Seq. Each researcher usually shares data analysis workflow through GitHub or as supplementary information of papers. Although it is helpful to prepare a detailed documentation, the authors could do that as part of GitHub document. In this case, there was no significant technological advancement or critical evaluation of the method. There is still much to specify by users, and it is unclear how this analysis workflow contributes to standardization of miRNA analysis.
Authors response) Thank you for providing us with feedback on our manuscript. We appreciate your suggestion to share our myBrain-Seq analysis workflow through GitHub or as supplementary information in papers. Although we recognize the importance of sharing a workflow, we believe that our manuscript offers much more than that. In our opinion, detailed documentation should not replace publications in peer-reviewed journals, especially for tools that are intended to be publicly available, such as myBrain-Seq.
MyBrain-Seq offers a comprehensive miRNA-Seq analysis tool, covering everything from raw data to functional analysis. We have made important adaptations to process miRNA data and ensure the replicability of results, minimizing the need for user intervention, based on our previous research (10.1007/978-3-030-86258-9_5, 10.1016/j.compbiomed.2021.104603). However, we acknowledge that the initial data preparation step is essential and cannot be skipped.
Upon thorough review of the article, we are confident that the contributions of myBrain-Seq are clearly stated. We have also included references to our previous research for a better understanding of this topic. If you have specific concerns, we kindly request that you bring them to our attention so that we can address them accordingly.
Reviewer 2 Report
The manuscript “myBrain-Seq: a pipeline for miRNA-Seq data analysis in neuropsychiatric disorders” by Daniel Pérez-Rodríguez et al offers an attractive tool for the analysis of miRNA expression data from RNA_Seq technology, which manages to bring together many of the most widely used tools in the field in a stand-alone solution. In particular, it allows the analysis from raw data to more in-depth analysis at the level of pathways of interest, functional enrichment, and network analysis in a guided manner that is quite intuitive.
Major points:
-
In section 2.6: it would be of interest the possibility to associate the clustering based on correlation (Pearson or Spearman Coefficient), if the starting sample size allows the evaluation of this statistical parameter. It has been shown that both measures (euclidean and correlation) are able to highlight different aspects of biological interest
-
In sections 3.3: alongside with the explanation for the null overlap of enriched pathways between the two strategies, it would be interesting to have a comparison of the ranking of the different enriched pathways. For instance: the more specific pathways found by myBrain-Seq are present in the complete significant list of pathways found by Pérez-Rodriguez et al 2023? If so, with which p- and q-value? In which position?
-
In section 4, lines 425-429: authors should deepen more the discussion of the new findings in the case-study reported
Minor points:
-
Section 2.7: the summary of the strategy proposed by Godard and van Eyll used for the functional analysis should be explained better
-
Fig 4 – it would be better to align the A-B-C-D parts on the left
Author Response
The manuscript “myBrain-Seq: a pipeline for miRNA-Seq data analysis in neuropsychiatric disorders” by Daniel Pérez-Rodríguez et al offers an attractive tool for the analysis of miRNA expression data from RNA_Seq technology, which manages to bring together many of the most widely used tools in the field in a stand-alone solution. In particular, it allows the analysis from raw data to more in-depth analysis at the level of pathways of interest, functional enrichment, and network analysis in a guided manner that is quite intuitive.
Major points:
- In section 2.6: it would be of interest the possibility to associate the clustering based on correlation (Pearson or Spearman Coefficient), if the starting sample size allows the evaluation of this statistical parameter. It has been shown that both measures (euclidean and correlation) are able to highlight different aspects of biological interest.
Authors response) We really appreciate this insightful suggestion, we agree with the reviewer and added this function to the new versión v1.1.0 of myBrain-Seq.
- In sections 3.3: alongside with the explanation for the null overlap of enriched pathways between the two strategies, it would be interesting to have a comparison of the ranking of the different enriched pathways. For instance: the more specific pathways found by myBrain-Seq are present in the complete significant list of pathways found by Pérez-Rodriguez et al 2023? If so, with which p- and q-value? In which position?
Authors response) We appreciate this remark. We added an explanatory text in section 3.3 with the requested comparison. We also added supplementary table 4 with a table comparing p-values, q-values and ranking position of each of the coincident pathways.
- In section 4, lines 425-429: authors should deepen more the discussion of the new findings in the case-study reported
Authors response) We agree with this comment and have included the main findings of the case study in that paragraph.
Minor points:
- Section 2.7: the summary of the strategy proposed by Godard and van Eyll used for the functional analysis should be explained better.
Authors response) We appreciate this suggestion. While we understand your concern, we believe that the paragraph in question provides the necessary information to grasp how the Godard & Van Eyll method is implemented in myBrain-Seq. The original publication (Godard, P. and Van Eyll, J. (2015)) goes into great detail on the complete method and its rationale.
- Fig 4 – it would be better to align the A-B-C-D parts on the left
Authors response) Thank you for your remark. We have reviewed figure 4 and corrected the alignment of the labels.
Reviewer 3 Report
The article is devoted to the description of a unified package for the analysis of miRNA sequencing data. This kind of work is very useful and allows for a comparative analysis of data. My comments:
1. Figure 1 should be reformatted.
2. Figure 4 should be compressed, reducing the accuracy to 3..5 characters.
3. Table 2 should be divided into two parts.
4. The results presented in Table 2 should be compared more carefully. Here, not only the categories are different, but also p- and q-values - the Perez-Rodriguez values are many orders of magnitude lower. I note that although the categories are different, there is much in common. For example, the Pérez-Rodriguez table has a category for "cell cycle" and the authors table has two categories related to cyclins.
5. The authors should justify the choice of tools used in the package. In particular, the choice of a tool for searching for targets.
Author Response
The article is devoted to the description of a unified package for the analysis of miRNA sequencing data. This kind of work is very useful and allows for a comparative analysis of data. My comments:
- Figure 1 should be reformatted.
Authors response) Thank you for your suggestion. However, after careful consideration, we have all agreed that Figure 1 effectively serves as a visual aid for comprehending the functionality of myBrain-Seq, and we have not identified any apparent issues with its format. In case you want, you may provide us with more specific details to help us better understand your concerns.
- Figure 4 should be compressed, reducing the accuracy to 3..5 characters.
Authors response) Thank you for your suggestion, and we apologize for any confusion. The purpose of Figure 4 is to display the results generated by myBrain-Seq software. If we were to reduce the number of decimals displayed in the figure, it would involve modifying the original results, which are intended to be consistent with the output provided by myBrain-Seq. Therefore, we believe that Figure 4 should maintain the same level of accuracy as that offered by myBrain-Seq.
- Table 2 should be divided into two parts.
Authors response) Thank you for your suggestion. After careful consideration, we believe that the current format of Table 2 facilitates easy comparison between the results of the case-study and those of myBrain-Seq. In our opinion, splitting this table into two sections would make it harder to understand. However, we made it easier to read by including horizontal lines between the rows.
- The results presented in Table 2 should be compared more carefully. Here, not only the categories are different, but also p- and q-values - the Perez-Rodriguez values are many orders of magnitude lower. I note that although the categories are different, there is much in common. For example, the Pérez-Rodriguez table has a category for "cell cycle" and the authors table has two categories related to cyclins.
Authors response) Thank you for your comment. We have addressed it by adding more detailed information in section 3.3. Additionally, we have included a supplementary table (table 4) that compares the p-values, q-values, and ranking positions of each of the coincident pathways.
- The authors should justify the choice of tools used in the package. In particular, the choice of a tool for searching for targets.
Authors response) Thank you for your feedback. We would like to clarify that the tools used in the myBrain-Seq software were selected based on a thorough review of the literature on miRNA-Seq analysis of neuropsychiatric data (10.1016/j.compbiomed.2021.104603), as detailed in section 3.3 of our manuscript. We took your suggestion and added our reasoning for choosing DIANA TarBase as our annotation database (lines 203-206).
Reviewer 4 Report
In order to be useful, tools of this type, which are basically based on a collection of other tools developed by third parties, must focus on making the analysis work really easy without making their installation and configuration more complex than simply using the original tools separately. The authors of myBrain-Seq have taken great care to ensure that this is the case by choosing a simple installation method and organizing the configuration files in an intuitive way.
It is to be welcomed that options are included to perform partial analyses (to or from a specific point) and to parallelize tasks, which makes it more convenient to tackle projects involving many samples. Along with the code, sample data and performance estimates on a standard machine are provided so that the end user can test the system and get an idea of how it will work on their own computer. I also think it is a good thing that the error checking does not interrupt the whole protocol unnecessarily, when these are partial errors that would allow the execution of the next steps. The user can check if there has been a problem by accessing .log files.
Regarding the tool itself, it consists of a collection of methods and algorithms well known and well appreciated by the scientific community (FastQC, Cutadapt, Bowtie, samtools, featureCounts, DESeq2, edgeR...). There are no surprises in this regard. Even when two alternatives are offered for one step of the analysis (DESeq2 and edgeR for the detection of differentially expressed miRNAs), it is possible to evaluate the intersection of the results obtained with each of them, facilitating the selection of results common to both.
As for not so positive aspects, I have to say that I have not found the option to modify the default values of the relevant results filtering thresholds (FDR, log2FC). While other options of the analysis such as account normalization methods may be convenient not to modify in a tool like this, the appropriate value of the final quantitative filters, which make the extraction of results more or less strict, may be different in each project. In this sense, if these values can be changed, it would be convenient to indicate how.
Finally, the analysis of prominent biological pathway terms (intelligently based on the presence of miRNAs and not genes, to avoid biases of pathways represented by many genes), although it provides the results of all relevant pathways, limits the construction of a network of miRNA-protein interactions to the most prominent pathway. However, sometimes the most useful functional information is not found in the top scoring pathway, but in the second, third, etc. and I have not found the option to construct other networks with them. An option to build miRNA-protein networks for the "n" most prominent biological pathways would be desirable.
As a final comment, myBrain-Seq seems too specific a name for a tool intended for general analysis of miRNAs. It implies that the tool is only for the analysis of brain data, but, in principle, given the algorithms and databases used, there is no reason for this.
Author Response
In order to be useful, tools of this type, which are basically based on a collection of other tools developed by third parties, must focus on making the analysis work really easy without making their installation and configuration more complex than simply using the original tools separately. The authors of myBrain-Seq have taken great care to ensure that this is the case by choosing a simple installation method and organizing the configuration files in an intuitive way.
It is to be welcomed that options are included to perform partial analyses (to or from a specific point) and to parallelize tasks, which makes it more convenient to tackle projects involving many samples. Along with the code, sample data and performance estimates on a standard machine are provided so that the end user can test the system and get an idea of how it will work on their own computer. I also think it is a good thing that the error checking does not interrupt the whole protocol unnecessarily, when these are partial errors that would allow the execution of the next steps. The user can check if there has been a problem by accessing .log files.
Regarding the tool itself, it consists of a collection of methods and algorithms well known and well appreciated by the scientific community (FastQC, Cutadapt, Bowtie, samtools, featureCounts, DESeq2, edgeR...). There are no surprises in this regard. Even when two alternatives are offered for one step of the analysis (DESeq2 and edgeR for the detection of differentially expressed miRNAs), it is possible to evaluate the intersection of the results obtained with each of them, facilitating the selection of results common to both.
Authors response) Thank you for providing such positive feedback regarding the usability and functionality of my Brain-Seq. We appreciate the time and effort you put into making precise and valuable suggestions. Your feedback has been very encouraging.
As for not so positive aspects, I have to say that I have not found the option to modify the default values of the relevant results filtering thresholds (FDR, log2FC). While other options of the analysis such as account normalization methods may be convenient not to modify in a tool like this, the appropriate value of the final quantitative filters, which make the extraction of results more or less strict, may be different in each project. In this sense, if these values can be changed, it would be convenient to indicate how.
Authors response) Thank you for your valuable suggestion. We have incorporated it into myBrain-Seq v1.1.0 as per your advice. Additionally, we have included instructions on how to use this new functionality in the README file. We also made a brief mention of this update in the article (lines 172-173) as per your suggestion. Thank you again for your input.
Finally, the analysis of prominent biological pathway terms (intelligently based on the presence of miRNAs and not genes, to avoid biases of pathways represented by many genes), although it provides the results of all relevant pathways, limits the construction of a network of miRNA-protein interactions to the most prominent pathway. However, sometimes the most useful functional information is not found in the top scoring pathway, but in the second, third, etc. and I have not found the option to construct other networks with them. An option to build miRNA-protein networks for the "n" most prominent biological pathways would be desirable.
Authors response) Thank you for your valuable suggestion. We had actually intended to incorporate this feature in the upcoming version of myBrain-Seq. Unfortunately, due to time constraints, we are unable to implement it within the peer review timeframe.
As a final comment, myBrain-Seq seems too specific a name for a tool intended for general analysis of miRNAs. It implies that the tool is only for the analysis of brain data, but, in principle, given the algorithms and databases used, there is no reason for this.
Authors response) Thank you for your insightful comment. Our objective is to further enhance myBrain-Seq by incorporating features specifically designed for its application in neuropsychiatric research. These features aim to link the outcomes of functional analysis with established neurological pathologies or gene variants. As an open source project, we welcome any collaboration or contribution towards achieving this goal.
Round 2
Reviewer 1 Report
It is understandable that the authors wish to keep the documentation as a form of published paper. If this fits to the jounral's scope, this manuscript can be publsihed in present form.
Reviewer 2 Report
After the round of revisions, in my opinion, the manuscript “myBrain-Seq: a pipeline for miRNA-Seq data analysis in neuropsychiatric disorders” by Daniel Pérez-Rodríguez et al is ready for publication.
Minor point:
-
Fig 4 – As a personal taste of spatial organization, it would be better to align the A-B-C-D parts on the left – mow it appears aligned on the right.